# AGMO Inhibitor Reduces 3T3-L1 Adipogenesis

**DOI:** 10.3390/cells10051081

**Published:** 2021-05-01

**Authors:** Caroline Fischer, Annett Wilken-Schmitz, Victor Hernandez-Olmos, Ewgenij Proschak, Holger Stark, Ingrid Fleming, Andreas Weigert, Manuela Thurn, Martine Hofmann, Ernst R. Werner, Gerd Geisslinger, Ellen Niederberger, Katrin Watschinger, Irmgard Tegeder

**Affiliations:** 1Institute of Clinical Pharmacology, Faculty of Medicine, Goethe University, 60590 Frankfurt, Germany; Ca.Fischer@med.uni-frankfurt.de (C.F.); wilken-schmitz@em.uni-frankfurt.de (A.W.-S.); geisslinger@em.uni-frankfurt.de (G.G.); e.niederberger@em.uni-frankfurt.de (E.N.); 2Institute of Pharmaceutical Chemistry, Goethe University, 60438 Frankfurt, Germany; victor.olmos@itmp.fraunhofer.de (V.H.-O.); proschak@pharmchem.uni-frankfurt.de (E.P.); 3Institute of Biochemistry I, Faculty of Medicine, Goethe University, 60590 Frankfurt, Germany; weigert@biochem.uni-frankfurt.de; 4Fraunhofer Institute for Translational Medicine and Pharmacology (ITMP), 60596 Frankfurt, Germany; thurnhanau@googlemail.com (M.T.); Martine.Hofmann@itmp.fraunhofer.de (M.H.); 5Institute of Pharmaceutical and Medicinal Chemistry, Heinrich Heine University Düsseldorf, 40225 Duesseldorf, Germany; stark@hhu.de; 6Institute for Vascular Signaling, Centre for Molecular Medicine, Goethe University, 60590 Frankfurt, Germany; Fleming@em.uni-frankfurt.de; 7Fraunhofer Cluster of Excellence for Immune-Mediated Diseases, 60590 Frankfurt, Germany; 8Institute of Biological Chemistry, Biocenter, Medical University of Innsbruck, 6020 Innsbruck, Austria; ernst.r.werner@i-med.ac.at (E.R.W.); katrin.watschinger@i-med.ac.at (K.W.)

**Keywords:** AGMO, compound screen, enzyme activity assay, tetrahydrobiopterin, 3T3-L1 mouse fibroblasts, adipocytes, macrophage polarization

## Abstract

Alkylglycerol monooxygenase (AGMO) is a tetrahydrobiopterin (BH4)-dependent enzyme with major expression in the liver and white adipose tissue that cleaves alkyl ether glycerolipids. The present study describes the disclosure and biological characterization of a candidate compound (Cp6), which inhibits AGMO with an IC50 of 30–100 µM and 5–20-fold preference of AGMO relative to other BH4-dependent enzymes, i.e., phenylalanine-hydroxylase and nitric oxide synthase. The viability and metabolic activity of mouse 3T3-L1 fibroblasts, HepG2 human hepatocytes and mouse RAW264.7 macrophages were not affected up to 10-fold of the IC50. However, Cp6 reversibly inhibited the differentiation of 3T3-L1 cells towards adipocytes, in which AGMO expression was upregulated upon differentiation. Cp6 reduced the accumulation of lipid droplets in adipocytes upon differentiation and in HepG2 cells exposed to free fatty acids. Cp6 also inhibited IL-4-driven differentiation of RAW264.7 macrophages towards M2-like macrophages, which serve as adipocyte progenitors in adipose tissue. Collectively, the data suggest that pharmacologic AGMO inhibition may affect lipid storage.

## 1. Introduction

Alkylglycerol monooxygenase (AGMO) is the so far only enzyme in mammalians that cleaves alkyl ether glycerolipids, such as the peroxisome derived abundant glycerophospholipids [1]. These lipids are unique in that they have an alkyl chain that is attached to the *sn-1* position of the glycerol backbone via an ether bond. Other glycerophospholipids have acyl chains attached via ester bonds [2]. Ether lipids are major components of cell membranes [3,4] and lipid raft microdomains [5] and are essential for the maintenance of plasma membrane integrity, fluidity, insertion of receptors, the recruitment of downstream signaling molecules [6] and building of cell-to-cell contacts [7]. In the brain, ether lipids are required for synaptic structures [8], myelination [9] and regulation of blood–brain barrier permeability [10]. Deregulations of ether lipid composition and metabolism have been associated with a number of developmental [11] and neurodegenerative disorders [12], with metabolic diseases such as diabetes [13], cancer [14,15] and susceptibility towards infections [16,17,18,19]. Hence, ether lipid homoeostasis is paramount for cells and organisms, and AGMO is so far unique in its ability to cleave the alkyl ether bond. In addition to membrane glycerolether lipids, platelet activating factor (PAF) and 2-arachidonolglycerol ether, also known as noladin ether, carry alkyl-ether bounds and are AGMO substrates in vitro [20,21]. Noladin ether is a precursor of the most abundant endocannabinoid, 2-arachidonoylglycerol, which is produced in peripheral fat cells and contributes to obesity [22,23]. Noladin ether itself is an agonist of the cannabinoid 2 receptors [24], whose deficiency leads to aging-associated obesity in mice [25].

AGMO is localized in the membrane of the endoplasmic reticulum (ER) and was, therefore, originally named transmembrane protein 195, TMEM195 [26], which gained some attention, because genome-wide association studies (GWS) identified polymorphisms in the intergenic region of diglycerol kinase beta (DGKB) and TMEM195 in association with type 2 diabetes and obesity [27,28,29,30,31]. Although the variants of DGKB/TMEM195 were not studied in terms of AGMO/TMEM195 expression or function, the region is considered as metabolic “risk gene”, and the GWAS suggest that AGMO is involved in human metabolic diseases.

AGMO needs the enzyme cofactor tetrahydrobiopterin (BH4) to be catalytically active. BH4 also serves as essential cofactor for biogenic amine hydroxylases and nitric oxide synthases [32]. Hence, the availability of BH4 is an important determinant for ether lipid breakdown, and competitive situations may arise from deregulations of the other BH4-dependent enzymes. In particular, the induction of inducible NOS (NOS2) in inflammatory conditions may lead to a 1000-fold increase in BH4 requirements.

AGMO-mediated alkyl ether cleavage results in a glycerol and a toxic aldehyde derivative, which is instable and rapidly converted to the corresponding fatty acid, through fatty aldehyde dehydrogenase (FALDH). Specific assays have been developed based on these features [33,34], but the complex nature of the assay still hinders large library compound screening to identify small molecule compounds activating or inhibiting AGMO. Owing to the lack of specific inhibitors, AGMO’s biological roles for in vivo ether lipid metabolism, and for metabolic and other diseases are still largely unknown. In agreement with the genetic association studies, it has been demonstrated that alkylglycerols promote adipocyte differentiation and development of obesity [35,36,37,38]. White adipose tissue expansion in obesity occurs through enlargement of pre-existing adipocytes (hypertrophy) and through formation of new adipocytes from precursor cells (adipogenesis), the latter resulting in high numbers of smaller adipocytes [39]. Interestingly, production of BH4 is increased in white and brown adipose tissue under inflammatory conditions [40] that mimic the pro-inflammatory conversion of visceral fat in obesity [41,42] suggesting that high BH4 reflects a high demand of AGMO under these conditions.

To assess the effects of AGMO inhibition, we did a compound screen based on structural similarities with BH4 to search for specific inhibitors. We were able to identify one candidate, which was subsequently tested in assays of cell proliferation, viability, morphology and adipocyte differentiation using 3T3 mouse fibroblasts, human HepG2 hepatocytes and RAW264 mouse macrophages as cell models. In addition, we assessed AGMO expression in visceral fat and liver in high-fat diet (HFD) fed mice.

## 2. Materials and Methods

### 2.1. Microsome Preparation

Male Sprague Dawley rats (24 weeks old) were sacrificed by CO_2_ intoxication and decapitation. Livers were excised and shock-frozen in liquid nitrogen and stored at −80 °C. For microsome preparation, 1.5 mL cold Tris/sucrose buffer pH 7.6 (0.1 M Tris-HCl, 0.25 M sucrose, 1 mM PMSF) per g in liver issue were added for homogenization on ice using a Sorvall mixer homogenizer. Homogenates were centrifuged at 3000× *g* for 10 min at 4 °C. Supernatants were transferred to a new tube and centrifuged at 13,000× *g* for 10 min at 4 °C. The crude extract was then submitted to centrifugation at 40,000× *g* for 1 h at 4 °C to pelletize the microsome fraction. The pellets were shortly washed and finally homogenized with storage buffer (0.1 M K_2_HPO_4_/KH_2_PO_4_ pH 7.5, 1 mM EDTA, 1 mM DTT). Protein concentrations were determined using Bradford reagent. Aliquots of microsomal protein were stored at −80 °C.

### 2.2. Alkylglycerol Monooxygenase Activity Assay and Inhibitor Screening

Alkylglycerol monooxygenase (AGMO) activity was assessed by using a pyrene-labeled alkylglycerol (1-O-pyrenedecyl-sn-glycerol) as substrate, which is converted to pyrenedecanal by AGMO. The chemical synthesis of the substrate is described in the Appendix A. In a second enzyme reaction, pyrenedecanal is converted by the fatty aldehyde dehydrogenase (FALDH) to the corresponding acid (pyrenedecanoic acid), which was detected via high-performance liquid chromatography (HPLC) as described previously [26,43]. The production of pyrenedecanoic acid correlates with AGMO activity. Negative controls (without BH4, buffer instead of protein extract) were run in parallel. AGMO activity assay conditions were optimized in terms of microsomal protein, BH4 and substrate concentration. Screening of 18 potential AGMO inhibitors (listed in Appendix A)was performed by using a 10 µL AGMO reaction mixture for triple determination of substrate, intermediate and product. The reaction mixture contained final concentrations of 100 mM Tris-HCl, pH 8.5, 1 mM dithioerythritol (DTE), 0.1 mg/mL catalase, 200 µM nicotinamide adenine dinucleotide (NAD), 200 µM nicotinamide adenine dinucleotide phosphate (NADPH), 100 µM BH4 stabilized in 0.5 mg/mL BSA (all Sigma-Aldrich, Darmstadt, Germany), 30 µM 1-O-pyrenedecyl-sn-glycerol, 0.8 µg/mL recombinant FALDH, 0.4 μg/mL recombinant QDPR (MyBioSource, Köln, Germany) and 25 µg/mL microsomal protein. The compound solution in DMSO or DMSO-vehicle (1 µL) was incubated with 3.75 µL of 1:500 diluted microsomal protein (70 µg/mL) for 10 min at room temperature. The reaction was initiated by adding 5.25 µL of master mix (containing cofactors, enzymes and the substrate) followed by incubation for 60 min at 37 °C in the dark. The enzyme reaction was stopped by adding 990 μL methanol. After centrifugation for 5 min at 16,000× *g*, 20 μL of the sample were injected to a Gemini NX C18, 5 µm particles, 250 × 4.6 mm column with a precolumn (Phenomenex, Aschaffenburg, Germany) using a Chromaster HPLC system with a fluorescence detector (Hitachi, Krefeld, Germany). Elution (flow rate 1.0 mL/min) was performed with a mixture of 95% acetonitrile (AcN) (*v/v*) and 5% H_2_O containing 0.1% trifluoroacetic acid (TFA) for 8 min with a column temperature of 40 °C. Pyrenedecanoic acid was detected after 5.4 min via its fluorescence (340 nm excitation and 367 nm emission, detection limit 0.04 ng/mL) and quantified by using the peak area under the curves (AUCs) relative to a standard curve of pyrenedecanoic acid (0.25–50 ng/mL).

### 2.3. Fatty Aldehyde Dehydrogenase Activity Assay

Determination of FALDH activity was performed by using the same reaction mixture and detection of pyrenedecanoic acid as described for AGMO activity assessment. Instead of AGMO substrate, 30 µM of pyrene-labelled FALDH substrate pyrenedecanal (PDA) was used. Production of pyrenedecanoic acid was determined in the presence of 10 or 100 µM BH4 and 300 µM compound 6 or DMSO vehicle control.

### 2.4. Phenylalanine Hydroxylase Activity Assay

Phenylalanine hydroxylase (PAH) activity was assessed as described previously [44]. Briefly, a rat PAH maltose fusion protein was derived from rat liver amplified PAH cDNA cloned in pmalC2 (NEB) and expressed in *E. coli*. The fusion protein was purified using an amylose column and elution with maltose. In order to assess PAH activity, 60 µg/mL recombinant rat PAH were preincubated with 200 µM phenylalanine for 10 min at 37 °C, and the reaction was started by adding a mixture of 24, 75 or 240 µM tetrahydrobiopterin (BH4) and 104 units/mL catalase (Serva, Heidelberg, Germany). After incubation for 20 min at 37 °C, the reaction was stopped by adding 10 µL of 1 M HCl. Ten microliters were then injected onto an RP-18 column (125 × 4 mm, Lichrosphere, 5 µm particle size; Merck) and eluted with 75 mM KH_2_PO_4_ buffer containing 20% (*v/v*) acetonitrile, 10% (*v/v*) methanol, 0.44 g/L SDS and 1.5 µM EDTA, pH 3.1 (adjusted with 1 M trichloroacetic acid), at a flowrate of 0.8 mL/min. Tyrosine was detected by fluorescence (excitation 285 nm, emission 325 nm) with a detection limit of 2.5 pmol (0.25 µM). Effects of compound 6 (2’N-dimethyl-6,7-dimethyl tetrahydropterin) were determined in 3 independent experiments with 0, 3, 10, 100, 300 and 1000 µM inhibitor (dissolved in DMSO) in the presence of 24, 75 and 240 µM BH4.

### 2.5. Nitric Oxide Synthase Activity Assay

RAW264.7 murine macrophages were cultured in VLE-DMEM medium (Biochrom) supplemented with 10% FCS at 37 °C, 5% CO_2_, 95% humidity. At about 80% confluency cells were stimulated for 24 h with 20 ng/mL LPS and 20 ng/mL murine IFNγ to stimulate the expression of inducible nitric oxide synthase (NOS2, iNOS). NOS activity was assessed using a colorimetric Nitric Oxide Synthase Activity Assay Kit (ab211083, Abcam, Heidelberg, Germany) according to the manufacturer’s instructions. Briefly, stimulated cells were washed 3 times in 1× PBS, harvested in NOS assay buffer supplemented with Protease Inhibitor Cocktail (Roche, Mannheim, Germany) and centrifuged for 10 min at 4 °C at 10,000× *g.* The supernatant was collected, and protein concentration was determined using a BCA-based assay with a BSA standard curve. Compound 6 (3, 10, 30, 100, 300, 1000, 3000, 10,000 µM) or DMSO as vehicle was mixed to the protein extract (50 µg) and incubated for 10 min at room temperature before adding a NOS master mix (containing cofactors, substrate and 10–12 µM BH4), incubation at 37 °C for 30 min and subsequent analysis of the colorimetric product at absorbance of 562 nm.

### 2.6. Culture and Differentiation of Mouse 3T3-L1 Preadipocytes

The cells 3T3-L1 were cultured in DMEM/F-12 medium supplemented with 10% FCS in 10 cm dishes at 37 °C, 5% CO_2_ and 95% humidity. For quantitative QRT-PCR, 10^5^ cells/well were seeded in a 6-well plate, for immunofluorescence analysis, 5 × 10^4^ cells/well were seeded in an 8-chamber culture slide and for quantification of Oil-Red O staining, 3000 cells/well were seeded in a 96-well plate. At 72 h, cells were incubated with BH4, 300 µM or 1 mM Cp6 or DMSO (0.1%, vehicle) for a further 48 h. Treatments were maintained throughout the differentiation. Differentiation was initiated by adding differentiation medium (+IDI) containing 0.5 mM methylisobutylxanthine (IBMX), 1 µM dexamethasone and 1 µg/mL bovine insulin (day 0). At day 2, IDI medium was replaced by DMEM/F-12 with 10% FCS containing 1 µg/mL insulin (INS). At day 4, INS medium was partly replaced with fresh medium, and at day 7, the medium was replaced by DMEM/F-12 with 10% FCS and incubation for further 3 days (washout, day 7–10). The washout was done with/without Cp6. Control cells were cultured and treated in parallel without IDI/without insulin. Cells were harvested or fixed at day 10.

### 2.7. RAW264.7 Cell Lines and Culture

Murine RAW264.7 macrophages with stable knockdown of endogenous AGMO (shAGMO) or transgenic expression of human AGMO+FLAG-tag (+huAGMO) were used as described [43] and listed in Appendix A. Vector control RAW macrophages carried a lentiviral construct with shRNA for knockdown of luciferase (shLUC) [43]. RAW macrophages were grown at 37 °C, 5% CO_2_ and 95% humidity in VLE-DMEM medium (Biochrom) supplemented with 10% FCS and 2–4 µg/mL freshly added selection antibody, puromycin. AGMO expression was assessed by quantitative real-time PCR (QRT-PCR). For stimulation, 10^4^ cells/cm^2^ were seeded in a 6-well plate and cultured for 24 h and, then, incubated for 48 h with 20 ng/mL LPS plus 20 ng/mL murine IFNγ or with 40 ng/mL murine interleukin-4 (IL-4) (all Sigma-Aldrich) or were left untreated (unstimulated control). For immunofluorescence analyses, RAW cells were seeded in 12-well plates on cover slips 12–24 h before fixation.

### 2.8. Culture and Free Fatty Acid Induced Cytotoxicity in HepG2 Cells

Human HepG2 hepatocytes were grown at 37 °C, 5% CO_2_ and 95% humidity in DMEM with high glucose and 10% FCS. Free fatty acid (FFA)-induced toxicity followed by quantitative assessment of lipid droplet formation and cytotoxicity was done in a 96-well plate and seeding of 10^4^ HepG2 in 100 µL DMEM supplemented with 10% FCS. After 24 h, medium was replaced by serum-free DMEM containing 1% BSA and 100 µM BH4, 100 or 300 µM Cp6 or vehicle (0.1% DMSO) and was incubated for further 24 h. In order to initiate FFA-induced cytotoxicity, cells were treated either with a cocktail containing DMEM supplemented with 10% FCS and a mixture of 750 µM FFAs (palmitic acid: oleic acid, 1:2) and 1% BSA or a treatment control mix containing DMEM with 10% FCS and 1% BSA. FFA media contained BH4, Cp6 or vehicle. Incubation with FFAs lasted for a maximum of 24 h, and toxicity was assessed via microscopy. After the respective incubation period, WST, SRB or Oil-Red O staining were performed (explained below).

### 2.9. Cytotoxicity Assays

Cells were seeded in 100 µL medium/well of a 96-well plate (3000 cells 3T3-L1; 10,000 HepG2 and RAW264.7). Different concentrations of Cp6 (0.1, 0.3, 1, 3, 10, 30, 100, 300, 1000 µM) or DMSO (0.1%) were added at 48 h for HepG2 cells, or at 24 h (low density) or 72 h (high density) for 3T3-L1 cells, and cells were incubated in the presence of Cp6 or vehicle for 24 h. Medium was then replaced with serum-free medium, 10 µL WST-1 reagent (Sigma) was added, and cells were incubated for 90 min at 37 °C. Formation of the formazan dye, which relies on the metabolic activity, was then measured via its absorbance in a multi-well spectrophotometer (TECAN Infinite F200 Pro) at 405 and 620 nm (reference). Subsequently, the reaction was stopped by adding 20 µL 50% trichloroacetic acid, TCA, (*w/v*) to a final concentration of 10% TCA for 1 h at 4 °C. Wells were washed with distilled water 7 times and were dried. To quantify cell mass, fixed cells were stained by adding 100 µL 0.4% (*w/v*) sulforhodamine B in 1% acetic acid for 30 min at room temperature under shaking. Wells were washed with 1% acetic acid 5 times, and cells were lysed by adding 250 µL 10 mM Tris/HCl pH 10.5 for 10 min at room temperature. Absorbance of sulforhodamine B at 540 nm was determined, which correlates with cell mass.

### 2.10. Oil-Red O Staining

For quantitative analysis of lipid droplets and lipid droplets in differentiated and undifferentiated 3T3-L1 cells and HepG2, cells were cultured as described above. At the respective end points, cells were washed with 1x PBS and fixed with 4% paraformaldehyde (PFA) for 20 min at room temperature. Oil-Red O staining was performed by adding 100 µL of 0.2% (*w/v*) Oil-Red O in 60% isopropanol per well for 30 min at room temperature. Wells were washed 5 times with distilled water, and cells were lysed by adding 50 µL isopropanol for 10 min at room temperature. Absorbance at 540 nm was determined using a multi-well spectrophotometer (TECAN Infinite F200 Pro) and correlates with the abundance of lipid droplets [45].

### 2.11. Cell Cycle Analysis Using Flow Cytometry

Cells were seeded in 10 cm dishes and subjected to IDI differentiation and treatment with Cp6 or vehicle for 24 to 48 h. Cells were harvested by trypsinization, fixed with 80% ethanol, washed, incubated for 5 min with 0.125% Triton-X-100, washed again and stained with propidium iodide in PBS containing 0.2 mg/mL RNAse A. Stained cells were analyzed by flow cytometry (FACS Canto II, Becton Dickinson, Heidelberg, Germany). The cell cycle distribution i.e., the percentage of cells in subG1, G0/G1 (2N), S and G2/M (4N) phase, was assessed using FLowJo V10.6 using the univariate algorithm, which is similar to the Watson Pragmatic algorithm [46]. The model assumes Gaussian distributions of the 2N and 4N populations (G0/G1 and G2/M) and then uses a subtractive function to reveal the S-phase population. The cells within one standard deviation of the 2N and 4N medians are subtracted from the data, and the remaining cells (S-phase) are fit to a polynomial function, which is convoluted with the Gaussian distributions of the 2N and 4N populations to form the complete model. 

### 2.12. Quantitative Real-Time PCR

Cells were washed 3 times with 1× PBS and harvested by scraping. Cell suspensions were centrifuged at 10,000× *g* for 5 min at 4 °C; PBS was discarded. Visceral fat and liver tissue of control and high-fat diet (HFD) fed mice were homogenized in TRI reagent by a Mixer Mill MM 400 using ceramic balls. Total RNA of tissues and cells was extracted according to standard procedures using TRI reagent. RNA was quantified via spectrometry (TECAN Infinite F200 Pro) and reverse transcribed using oligo-dT and random hexamers as primers to obtain cDNA fragments (Verso cDNA Synthesis Kit, Thermo Fisher Scientific, Schwerte, Germany). Quantitative real-time PCR (QRT-PCR) was performed on an ABI 7500 Fast Real-time PCR System (Applied Biosystems) using the SYBR green technique (Maxima SYBR Green/ROX qPCR Master Mix or SYBR Select Master Mix, Thermo Fisher Scientific). Cycle numbers were normalized to a house-keeping gene (eukaryotic translation elongation factor 2, Eef2 or protein phosphatase, Ppa1), and transcript regulation was assessed relative to control/vehicle conditions using the comparative threshold cycle method (ΔΔCT) according to manufacturer’s instructions (Applied Biosystems). A melt curve analysis was performed to assess product specificity. Primer sequences are summarized in Appendix A.

### 2.13. Immunofluorescence

Cells were incubated and treated in slide culture chambers up to the respected endpoint. After washing with PBS, cells were fixed with 4% PFA. After permeabilization with 0.1% Triton-X-100, unspecific epitopes were blocked using 3% BSA in 1× PBS for 75 min at room temperature and subsequently incubated overnight with the primary antibody in 1% BSA in PBS at 4 °C. After washing, cells were incubated with the secondary antibody coupled to the respective fluorophore for 2 h at room temperature (antibodies in Appendix A). To visualize cell nuclei, 4′,6-diamidine-2-phenylindole (DAPI), staining was performed for 10 min in PBS, and slides were mounted in Aqua Poly/Mount medium. For microscopy, an inverted fluorescence microscope equipped with AxioVision 4.9 software (Zeiss, Jena, Germany) or an BZ-9000 inverted fluorescence microscope (Keyence, Neu-Isenburg, Germany) with automated image stitching function were used.

### 2.14. Animals

C57Bl/6J were used and had free access to food and water and were housed in climate and light controlled quiet rooms with a 12 h light–dark cycle. General well-being was assessed by daily inspections and monitoring of body weights (1–2/week). The experiments were approved by the local Ethics Committee for animal research (Darmstadt, Germany), adhered to the guidelines of the Society of Laboratory Animals (GV-SOLAS) and were in line with the ARRIVE guidelines and the European and German regulations for animal research. Male 24-week-old C57Bl/6J mice (*n* = 6) were fed with a high-fat Western-type diet (ssniff, Soest, Germany, diet TD88137) for 16 weeks. Control 22–23 weeks old C57Bl/6J mice (*n* = 6) were fed with standard food pellets in parallel. Mice were sacrificed by CO_2_ intoxication and cardiac puncture, and the liver and visceral fat were excised and immediately shock-frozen in liquid nitrogen. Tissue samples were stored at -80 °C until RNA extraction.

### 2.15. Statistics

Data are presented as mean and standard deviation (SD). Statistical analyses were done with GraphPad Prism 8. To assess enzyme activity, log-transformed concentrations of substrate or cofactor were plotted versus the relative change in product concentrations, and Km values were fitted according to standard Michaelis–Menten kinetics after normalization of raw enzyme activity data to percentages of the reference level (DMSO control). IC50 values of Cp6 for AGMO, NOS and PAH activity were determined by using standard Emax models. Group data were compared by parametric or non-parametric statistics according to subgroup structure and data distribution. Unpaired t-tests were employed for comparison of two groups of normally distributed data, and univariate or multivariate analyses of variance (ANOVA) were employed for comparisons or more than groups or interactions of parameters. ANOVA was followed by *t*-tests using an adjustment of alpha according to Šidák or according to Dunnett. Adjusted *p*-values are reported. Mann–Whitney U or Kruskal–Wallis tests were used in case of violations of normal distribution or low sample sizes of *n* = 3. Gauss or log-Gauss distributions were assessed with the Anderson Darling test.

## 3. Results

### 3.1. AGMO Activity Assay and Compound Screen

The first set of experiments focused on the establishment of a quantitative enzyme activity assay for AGMO using 1-pyrenedecylglycerol as substrate (chemical synthesis: Suppl. Methods, Appendix A). Exemplary HPLC chromatograms show the chromatographic separation of 1-pyrenedecylglycerol (substrate) and the product 1-pyrenedecanoic acid (Figure 1A). The log2 areas under the peaks (AUCs) were linear over a calibration range of 0.125 to 64 ng/mL of the product (Figure 1B1). AGMO activity was BH4-dependent and reached a plateau with low variability at about 100–250 µM of the cofactor (Figure 1B2,B3) and 25–100 µg/mL of endosomal protein. Subsequent experiments were done with 25 µg/mL protein and 100 µM BH4. At these conditions, maximum AGMO activity was reached with 30–100 µM of the substrate (Figure 1B4). Consequently, 30 µM of 1-pyrenedecylglycerol was used for screening of putative inhibitory effects of candidate compounds (chemistry and structures in Appendix A). The selection was based on their structural similarity to BH4. Hence, the aim was to target the BH4 binding site [47] and competitively displace BH4 without inherent cofactor function. The initial screen at high concentrations revealed five putative candidates (Figure 1C upper), and subsequent tests at decreasing concentrations identified complete inhibition of AGMO with compound 6 (Cp6; 2-(dimethylamino)-6,7-dimethyl-5,6,7,8-tetrahydro-4(3H)-pteridinone) (Figure 1C lower). Cp6 concentration versus effect analysis (Figure 1D1) showed a concentration-dependent sigmoidal inhibition of AGMO with half-maximum inhibition at about 50 µM (IC50 ranging from 26 to 100 µM). The IC50 for phenylalanine hydroxylase (PAH), which is one of the BH4-dependent hydroxylases, was about 5-fold higher (IC50 360–430 µM, Figure 1D3), whereas inhibition of NOS activity in RAW264.7 macrophages (mainly NOS3) required about 20-fold higher concentrations of Cp6 (IC50 1.9–5.3 mM, Figure 1D2). The 2nd-step enzyme in the activity assay, fatty aldehyde dehydrogenase (FALDH), which is required for conversion of the intermediate aldehyde, was not inhibited (Figure 1D4).

The results show that Cp6 preferentially inhibits AGMO over other BH4-dependent enzymes with 5–20-fold stronger inhibition of AGMO at IC50 concentrations, albeit with moderate potency. Hence, the next set of experiments tested the dose dependent cytotoxicity and cell biology of AGMO inhibition with Cp6 in three cell models.

### 3.2. Cp6 Inhibits Adipogenesis in 3T3-L1 Cells

Because of the importance of alkylglycerols for adipogenesis, AGMO’s high expression in the liver and white adipose tissue and AGMO-dependent polarization of macrophages [43], we opted for three cell models, namely adipogenesis from 3T3-L1 cells (Figure 2, Figure 3 and Figure 4), cytokine-driven polarization of RAW264.7 mouse macrophages towards M1-like and M2-like cells (Figure 5) and fatty acid-evoked toxicity in HepG2 hepatocytes (Figure 6).

Cp6 had no growth-inhibitory or cytotoxic effects in high-density cultures of 3T3-L1 cells. In low-density cultures, Cp6 reduced cell viability and proliferation as assessed via sulforhodamine B (SRB) and WST assays with half-maximum effect at 300–400 µM, which is about 5-fold higher than the IC50 for AGMO inhibition (Figure 2A), and Cp6 completely inhibited the differentiation of 3T3-L1 preadipocytes towards adipocytes using a standard protocol of “IBMX-DEX-Insulin” (IDI). Proliferation and differentiation was almost completely suppressed in the presence of Cp6 at day 3–10 of the differentiation process as compared to vehicle-treated cultures (Figure 2B). The inhibition of proliferation and cell growth was reversible if the Cp6-containing medium was replaced with vehicle medium at 7d. However, re-proliferating cells (during wash-out, day 7–10) still did not accumulate lipid droplets typical for adipocytes, which were abundant at 10 days in vehicle-treated cultures (Figure 2B).

Time course experiments (Figure 3A schedule; Figure 3C culture images) show a complete suppression of adipocyte proliferation and differentiation upon Cp6 treatment. The absence of adipocyte differentiation also manifested with unaltered expression of the preadipocyte factor 1, Pref-1 (alias Dlk1, protein delta homolog 1), but lack of expression of the mature adipocyte marker fatty acid-binding protein, FABP4 (Figure 3C) and absence of lipid droplets as assessed by Oil-Red O (ORO) staining (Figure 3B). The experiment was replicated with 300 µM Cp6 (Appendix A). The lower dose reduced the Pref-1 to FABP switch and ORO staining, but cells were in part hypertrophic. Interestingly, vehicle or BH4-treated cultures showed a strong increase in AGMO and endothelial NOS (eNOS/NOS3) expression (at mRNA level) upon differentiation with IDI (Figure 3D), suggesting that both enzymes plus BH4 were required for the differentiation and proliferation of adipocytes. Cp6-treated cells did not show such an increase in AGMO and eNOS expression. They also did not increase PPARγ and CEBPα expression (Figure 3D), which are key pro-adipogenic transcription factors.

To assess the step at which Cp6 interrupted adipocyte proliferation and adipogenesis, we analyzed the mitotic clonal expansion as the first and crucial step of adipocyte differentiation via flow cytometry-based cell cycle analysis (Figure 4A–C). Mitotic clonal expansion manifested in an increase in S-phase and G2/M cells 24 h after starting the IDI stimulation (Figure 4A top). The increase in S-cells and of G2/M cells was stronger in Cp6 (300 µM)-treated cells (Figure 4A bottom). G1 cells were reduced, and a relevant proportion shifted to sub-G1 cells, which are apoptotic/dead cells (Figure 4A,B). The cell cycle distribution suggested an S to G2/M block under Cp6, leading to hypertrophy of existing cells and premature cell death. In a second independent experiment (Figure 4C), we assessed the time course of clonal expansion. Again, there were more cells in the S-phase in the Cp6 group at 1d after starting IDI and more cells in G2 at 3d suggesting a slowing of the cell cycle. In agreement with the alterations of the cell cycle, Cp6 increased cyclin D expression, which is needed for G1 to S-transition, but reduced cyclin A needed for S to G2/M transition (Figure 4D).

### 3.3. Cp6 Does Not Affect Macrophage Proliferation but Blocks the M1 to M2 Polarization

The obesogenic hyperplasia of white adipose tissue is associated with a pro-inflammatory response and metabolic disease, and AGMO was shown to drive polarization of macrophages towards an M2-like phenotype, which contributes to the resolution of inflammation and is protective in the context of inflammation but permissive in the context of cancer growth [48,49]. It is also a niche for adipocyte progenitors [50,51]. In the first set of experiments, we confirmed that shAGMO RAW264.7 cells (knockdown) had low expression, and transgenic +huAGMO RAW264.7 (overexpression of human AGMO) had a high expression of AGMO at mRNA (Figure 5A) and activity levels (Figure 5B). As expected, AGMO activity increased upon M2 polarization evoked with IL-4 but dropped upon M1-like polarization evoked with LPS/IFNγ (Figure 5B, right). SRB and WST absorbances were substantially lower in shAGMO RAW cells as compared to shLUC (control) and +huAGMO cells suggesting a lower metabolic rate (Figure 5C). As expected, Cp6 had no additional deleterious effect on the viability, whereas it moderately reduced SRB and WST readouts in shLUC and +huAGMO cells, albeit only at high concentrations (Figure 5C). However, Cp6 at 100 µM strongly reduced IL-4 induced M2 polarization in +huAGMO RAW264.7 macrophages and moderately fortified the occurrence of M1-like markers, as assessed by the analysis of each four typical M1 and M2 markers at mRNA level (Figure 5D,E). Hence, Cp6 treatment had similar effects as AGMO knockdown on macrophage polarization.

### 3.4. AGMO Is Upregulated in the Liver under HFD but Cp6 Does Not Protect against Free Fatty Acid-Induced Toxicity in Hepatocytes

To assess the putative in vivo relevance of obesogenic AGMO expression, mice were fed with a high-fat Western-type diet with 42% calories from fat for 16 weeks (Appendix A). AGMO expression was increased in the fatty liver in obese mice but was reduced in gonadal white adipose tissue (Appendix A), possibly owing to a predominant WAT hypertrophy rather than adipogenesis at this site. In parallel, eNOS expression was reduced in HFD fed mice in fat tissue (liver n.s.), in line with previous studies [52,53].

High AGMO expression in the liver under HFD suggested that Cp6 might protect hepatocytes from free fatty acid (FFA)-evoked cytotoxicity by preventing fat accumulation, which is a model for liver steatosis [54,55]. Therefore, we tested Cp6 in hepatocytes exposed to FFAs. Cp6 had per se no effect on HepG2 cell viability up to high concentrations (Figure 6A,B). Cp6 did not prevent the FFA-evoked drop of HepG2 viability and proliferation (SRB and WST assays), but it significantly reduced the FFA-evoked accumulation of lipid droplets as identified by Oil-Red O at 24 h of FFA exposure (Figure 6C,D). Hence, Cp6 had some protective effect in this in vitro model of non-alcoholic liver steatosis.

## 4. Discussion

The present study describes the disclosure of a compound (Cp6) that inhibits AGMO activity with moderate potency and moderate selectivity for AGMO over other BH4-dependent enzymes. Owing to its chemical and structural similarity to BH4, we assume that it competes with BH4 for the cofactor binding site but is not able to function as a coenzyme for AGMO. IC50 levels for phenylalanine hydroxylase (PAH) and NOS were higher than for AGMO, suggesting lower affinity to the BH4 binding sites of these enzymes. The relative specificity was the prerequisite to study the biological effects of AGMO inhibition.

Previous studies suggested that AGMO expression increases upon IL-4-driven differentiation of macrophages towards M2-like cells [43], which was confirmed in our studies in RAW cells. Hence, we expected that its inhibition would block the differentiation, and indeed, Cp6 blocked the IL-4-stimulated transcriptional activation of M2-like marker genes. M2-like macrophages constitute a niche for adipocyte precursor cells in adipose tissue [50,51], promote angiogenesis and are permissive for cancer growth [56,57]. Because of these features and in vivo transcriptional regulation of AGMO on high-fat diet, we studied Cp6 effects on adipogenesis. AGMO expression increased upon differentiation of 3T3-L1 cells towards adipocytes, and Cp6 halted the proliferation and lipid droplet accumulation, suggesting that AGMO supported the generation of newborn adipocytes and storage of lipids in lipid droplets. It is of note that alkylglycerols (AKG) in breast milk were shown to impede the transformation into lipid-storing white adipose tissue (WAT) during infancy [58]. Breast milk AKGs were metabolized by adipose tissue macrophages towards the alkylglycerol ether, platelet-activating factor (PAF), which favorably maintained beige adipocytes during infancy [58]. PAF is an AGMO substrate [20,21] suggesting that inhibition of AGMO may alter the balance of AKG and AKG-ethers, including PAF.

At 300 µM, Cp6 favored the occurrence of large adipocytes and moderately reduced lipid droplets suggesting that AGMO inhibition may result in hypertrophy of existing cells. WAT hyperplasia results from hypertrophy of pre-existing adipocytes and from proliferation and differentiation from precursor cells [39]. Newly formed small adipocytes appear to be metabolically less unfavorable and less pro-inflammatory than their hypertrophic congeners [39]. In support, activation of PPARg by thiazolidinediones results in an increase in the number of small adipocytes and improves glucose metabolism [59,60]. We show that Cp6 impairs adipogenesis as revealed by reduced lipid droplet accumulation and reduced PPARγ expression. Hence, treatment with Cp6 may lead to fewer small adipocytes but rather an increase in the number of hypertrophic cells, which may have negative effects on the pro-inflammatory environment in obese adipose tissue. In line, the observation of reduced AGMO expression found in white adipose tissue of HFD-fed mice may suggest that reduced AGMO is a marker of de-differentiation of adipocytes, suggesting that inhibition of AGMO using Cp6 might have negative consequences on visceral fat and associated diseases.

On the other hand, Cp6 inhibited the accumulation of lipid droplets in adipocytes and hepatocytes, and adipocytes maintained low numbers of lipid droplets when Cp6 was removed suggesting that AGMO-driven lipid metabolism promotes the formation or life cycle of lipid droplets and storage of triglycerides. Biogenesis of lipid droplets originates from ER membranes [61] where AGMO is localized. AGMO inhibition might reduce the expansion of visceral fat, if such antiproliferative effects were to occur in vivo. In a previous study, lipid droplets were absent in ether-lipid-deficient mice and were restored with an AKG-rich diet that also alleviated the progression of the pathology of these mice in testis and adipose tissue [62], both sites with high AGMO expression [63].

With Cp6, we have now a tool to study if AGMO is involved in lipid droplet formation or expansion and can elucidate the observed effects in adipocytes and hepatocytes and the inverse regulation of AGMO expression in WAT and liver in vivo in high-fat diet fed mice. We expected an increase in AGMO expression under these conditions, but in contrast to the liver, where AGMO was highly upregulated, we found a decrease in AGMO expression in visceral WAT in fat mice. This appears to contradict the observed upregulation of AGMO in proliferating adipocytes in vitro and possibly points to a high contribution of hypertrophy of existing adipocytes in vivo at this site.

## 5. Conclusions

The study shows that Cp6 is a weak inhibitor of AGMO with some specificity and low toxicity. The data suggest that treatment with Cp6 may affect adipogenesis and lipid storage-associated diseases. Suppression of adipogenesis may be favorable but might drive hypertrophy of existing adipocytes with pro-inflammatory outcome. Further studies of the molecular mechanisms and adaptations upon AGMO inhibition and in vivo studies are needed to reveal AGMO’s putative function for lipid-associated diseases.

## Figures and Tables

**Figure 1 cells-10-01081-f001:**
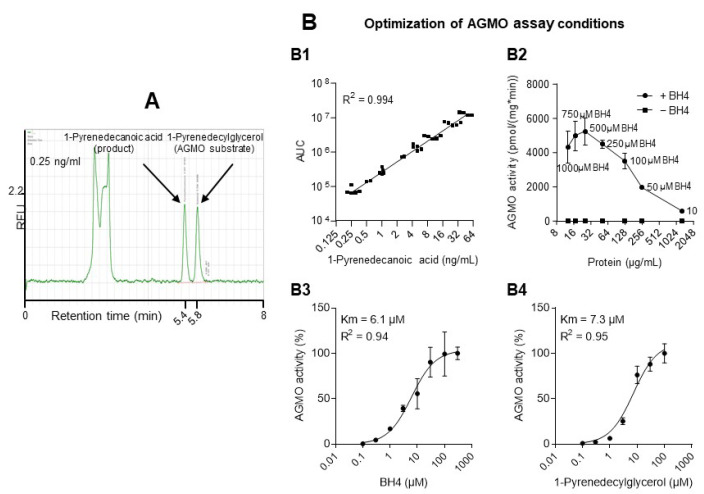
Enzyme activity of alkylglycerol monooxygenase (AGMO) and compound screening. (**A**) Exemplary HPLC chromatograms show the fluorescent 1-pyrenedyclglycerol AGMO substrate and the product 1-pyrenedecanoic acid which is detected at 5.4 min via its fluorescence (340 nm excitation and 367 nm emission). Chromatograms show efficient separation of substrate and product at 0.25 ng/mL substrate concentrations. (**B**) Optimization of AGMO activity assay conditions. (**B1**) Association of substrate concentrations with product peak areas (areas under the curve, AUCs). (**B2**) AGMO enzyme activity in the presence and absence of the coenzyme, tetrahydrobiopterin (BH4 in µM) in dependence of microsome protein concentration. (**B3**) Michaelis–Menten kinetics with Km values depending on concentrations of its cofactor BH4. (**B4**) Michaelis–Menten kinetics with Km values depending on substrate concentrations of 1-pyrendyclglycerol follows. Data points represent the means with SD of 3 experiments. (**C**) Compound screen of 18 candidate AGMO-inhibitory compounds. AGMO activity was assessed at 25 µg/mL microsomal protein, 100 µM BH4 and 30 µM 1-pyrenedecylglycerol. Candidate compounds were selected according to their structural similarity to BH4. The screening revealed 5 putative candidates that led to a decrease >80% of AGMO activity (6, 12, 13, 14, 17). Cp6 showed the strongest effect. The data represent the means ± SD of the of relative AGMO activity of *n* = 3 experiments. (**D**) Assessment of Cp6 inhibitory effects on AGMO and of putative unspecific inhibitory effects on phenylalanine hydroxylase (PAH), nitric oxide synthase (NOS) and free fatty aldehyde dehydrogenase (FALDH). (**D1**) Cp6 inhibited AGMO in a BH4-independent manner with an IC50 of 26–100 µM (*n* = 3). (**D2**) Cp6 inhibited NOS activity from LPS, and IFNγ stimulated RAW macrophages at IC50 of 1900–5300 µM, i.e., 10–20-fold higher than the IC50 for AGMO (*n* = 3). (**D3**) Cp6 inhibited PAH activity in a BH4-independent manner at high IC50 values of 360–430 µM (*n* = 3). (**D4**) Cp6 significantly inhibited AGMO activity (* *p* = 0.0317, ** *p* = 0.0067), but Cp6 had no effect on FALDH, the second essential enzyme of the AGMO activity assay. Bars show means with SD.

**Figure 2 cells-10-01081-f002:**
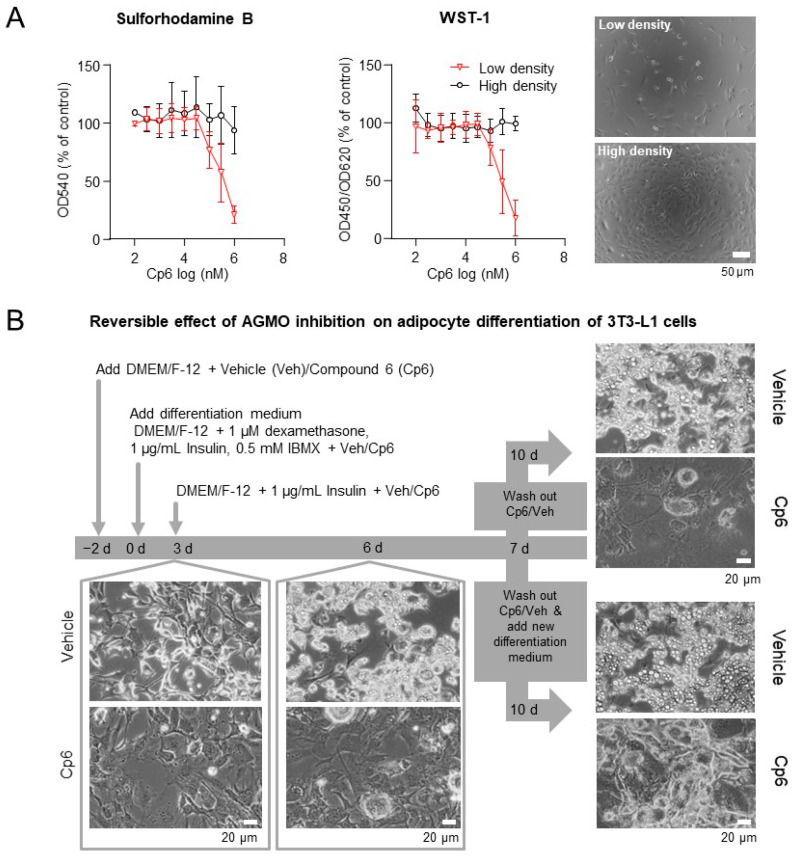
Cp6 reversibly inhibits differentiation of 3T3-L1 preadipocytes. (**A**) Effect of Cp6 on cell viability and proliferation as assessed by sulforhodamine B (SRB) and WST-1 assays in low-density and high-density cultures of undifferentiated 3T3-L1 mouse fibroblast-like cells (exemplary culture images right). Data are means ± SD (*n* = 2–5 independent experiments). Cp6 reduced viability and proliferation in low-density cultures (IC50 300–400 µM, best-fit value of pooled data) but had no impact on cell viability at high culture density. Scale bar 50 µm. (**B**) Flowchart of the differentiation of 3T3-L1 fibroblast-like cells towards adipocyte-like cells in the presence of Cp6 (1 mM) or vehicle (0.1% DMSO) up to 6d of continuous treatment, and subsequent “wash-out” from day 7–10. Exemplary culture images show 3T3-L1 preadipocytes at different stages of the differentiation process. Cp6 reversibly inhibited the proliferation and irreversibly prevented the formation of lipid droplets. Scale bars 20 µm.

**Figure 3 cells-10-01081-f003:**
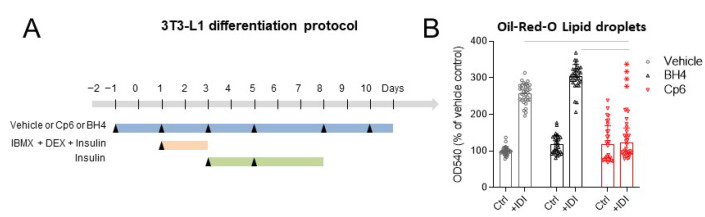
Cp6 preserves preadipocyte markers and prevents accumulation of lipid droplets. (**A**) Flowchart showing the differentiation procedure with/without treatment of 3T3-L1 preadipocytes. Cells were treated with vehicle (0.1% DMSO), 1 mM Cp6 or 100 µM BH4 before and continuously after addition of differentiation medium (+IDI). The differentiation medium (IDI+) consisted in dexamethasone, 3-isobutyl-1-methylxanthine (IBMX) and insulin, referred to as IDI+. Controls were treated in parallel without differentiation medium (-IDI). (**B**) Quantification of lipid droplets in vehicle, Cp6 (1 mM) or BH4 (100 µM)-treated 3T3-L1 preadipocytes after 10 days of differentiation. Lipid droplets were stained with Oil-Red O and absorbance read at 540 nm and normalized to non-differentiated vehicle-treated controls. Data were compared with 2-way ANOVA and subsequent post hoc analysis for “treatment”. The scatter show replicates of *n* = 5 independent experiments, bars and whiskers show means ± SD, **** *p* ≤ 0.0001. (**C**) Morphology and expression of pre/adipocyte markers in 3T3-L1 pre/adipocytes after 10d of culture with/without differentiation medium (+IDI, –IDI) in the presence of vehicle (0.1% DMSO), 1 mM Cp6 or 100 µM BH4. Differentiating cells (IDI+ cultures) receiving vehicle or BH4 started to develop lipid droplets from 3 days on. Lipid droplets did not develop in Cp6-treated cells. Final staining with Oil-Red O (ORO) solution (at 10d) confirmed the complete absence of lipid droplets in Cp6-treated cultures. Vehicle and BH4-treated 3T3-L1 cells expressed fatty acid binding protein (FABP4), which is a marker for mature adipocytes, whereas Cp6-treated cells preserved the preadipocyte marker expression of preadipocyte factor 1 (Pref-1, gene Dlk, protein delta homolog 1). (**D**) QRT-PCR analysis of AGMO, eNOS, PPARγ and CEBPα mRNA expression in 3T3-L1 pre/adipocytes. Eukaryotic elongation factor 2 (Eef2) was used as a housekeeping gene, and mRNA was normalized to non-differentiated vehicle control cultures set to 1. Data were compared with 2-way ANOVA and subsequent post hoc Dunnett versus vehicle control; *n* = 5–6 independent cultures, bars and whiskers show means ± SD. The asterisks show significant results versus “vehicle +IDI”; **** *p* <0.0001, *** *p* < 0.001.

**Figure 4 cells-10-01081-f004:**
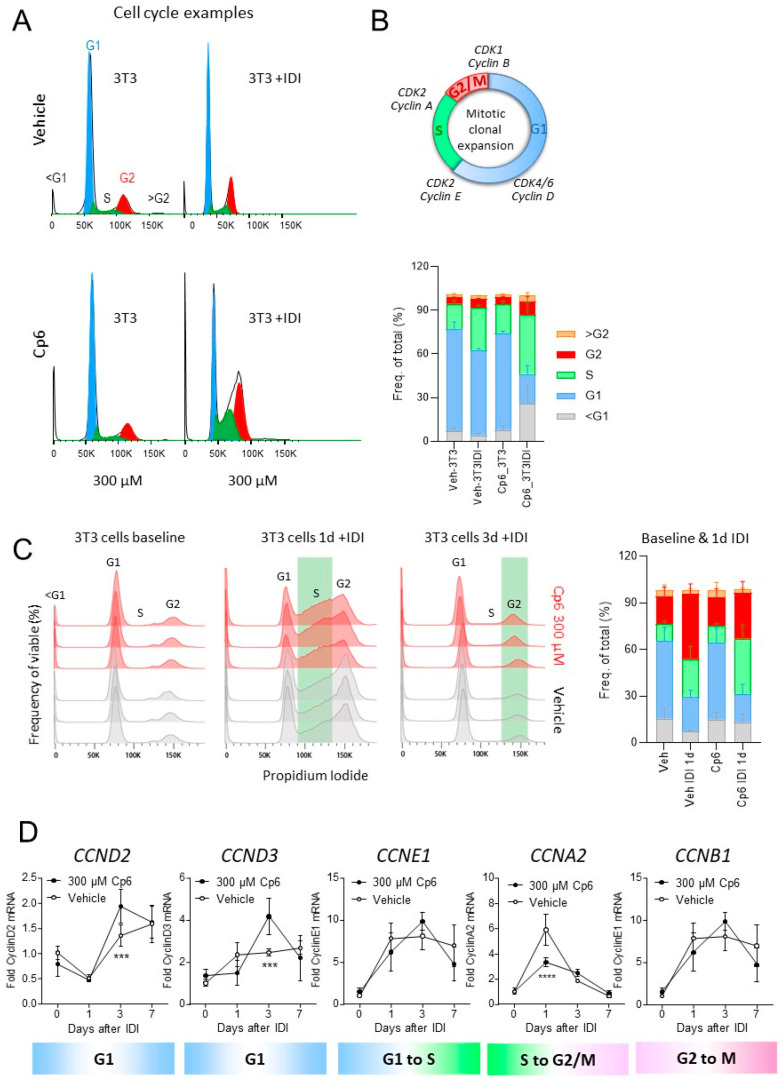
Cp6 slows down cell cycle progression of 3T3 adipocytes during IDI-mediated clonal expansion. (**A**) Exemplary histograms of the cell cycle distribution as assessed per flow cytometry analysis of the DNA content via propidium iodide staining. The 3T3 cells were treated with vehicle (upper), or 300 µM Cp6 (bottom) with/without IDI differentiation medium for 24 h. One × exp5 cells were counted per culture, *n* = 5 per condition, and propidium iodide intensities were fitted according to the Watson Pragmatic model implemented in FlowJo. (**B**) Drawing of the cell cycle and major cyclins involved in transitions and quantitative result of the cell cycle distribution as shown in A. Cells in the subG1 phase (<2N) are considered apoptotic/dying cells. Cells with >4N DNA are considered polyploid. Data were compared via 2-way ANOVA for “IDI” and “treatment” and subsequent *t*-tests with Šidák adjustment of alpha. (**C**) Time course of the cell cycle distribution in 3T3 cells at baseline, 1d and 3d after adding IDI. Cells were treated with vehicle or 0.3 mM Cp6 (*n* = 6–9 per group and time point). At 1d, there are more Cp6-treated cells in S, at 3d, there are more in G2. Two-way ANOVA for the interaction “CC-phase X group” reveals a significant difference between groups, *p* < 0.0001 at 1d, *p* = 0.0103 at 3d. (**D**) Analysis of cyclin expression on mRNA level during adipogenesis in 3T3-L1 cells. Data were compared with 2-way ANOVA for “time” x “treatment” and subsequent post hoc analysis for “treatment” using a correction of alpha according to Sidak versus vehicle control; *n* = 6 independent cultures, bars and whiskers show means ± SD, *** *p* < 0.001.

**Figure 5 cells-10-01081-f005:**
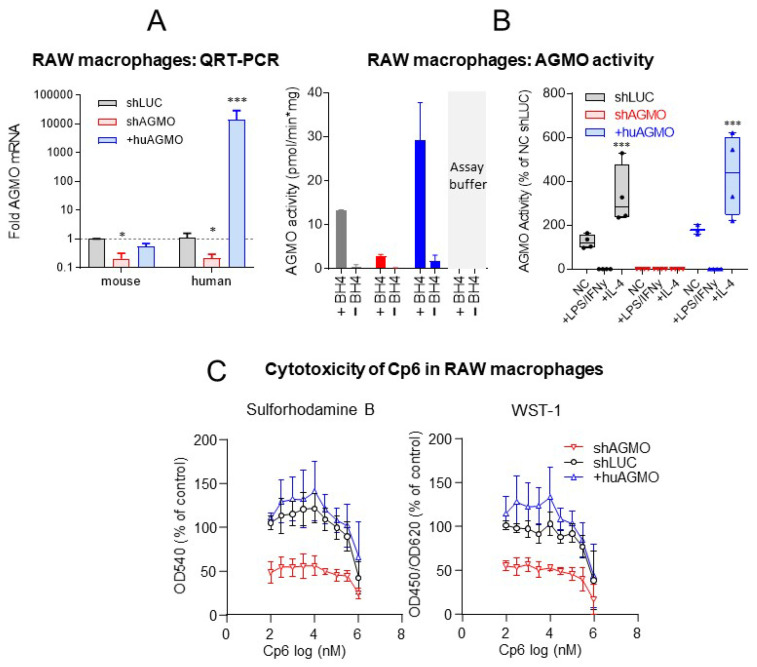
Cp6 inhibits macrophage M1 to M2 polarization. (**A**) Relative mRNA of endogenous murine and transgenic human AGMO in shAGMO and in +huAGMO RAW 264.7 macrophages relative to control cells (shLUC). Bars show means ± SD. Data were compared with one-way ANOVA separately for murine and human AGMO expression and subsequent post hoc analysis using a correction of alpha according to Dunnett versus control cells; *n* = 3–6, *** *p* < 0.001, **** *p* < 0.0001). (**B**) AGMO activity in homogenates of in shAGMO and in +huAGMO RAW264.7 macrophages and shLUC RAW control cells. (NC, naïve control). Bars and whiskers show means ± SD (*n* = 2–3). In the right panel, cells were stimulated with LPS+IFNγ (20 ng/mL each) or with IL-4 (40 ng/mL) (*n* = 3–4 per group and stimulus). The box represents the interquartile range, the line is the median, and the whiskers show minimum and maximum. The scatters show results of individual samples. Data were compared with 2-way ANOVA for “cell line” by “stimulation” using a correction of alpha according to Dunnett (*n* = 3–4; * indicates differences between cell lines. (**C**) Effect of Cp6 on cell viability (sulforhodamine B assay, SRB) and cell proliferation (WST-assay) in shLUC, shAGMO and +huAGMO RAW264.7 macrophages. Data points show means with SD (*n* = 6–9 per cell line of 3 independent experiments) as percentage of treated control RAW264.7 cells. (**D**) Relative mRNA expression of M1 polarization markers (IL23a, IL6, NOS2/iNOS, TNFalpha) in vehicle or Cp6 (100 µM)-treated control and LPS+IFNγ (20 ng/mL each) stimulated +huAGMO RAW264.7 macrophages. Data were compared with 2-way ANOVA for “stimulation” x “treatment” and subsequent post hoc analysis using a correction of alpha according to Tukey. Asterisks indicate significant differences between vehicle and Cp6 (*n* = 5, * *p* < 0.05, ** *p* < 0.01). Bars and whiskers show means ± SD. (**E**) Relative mRNA expression of M2 polarization markers (Tgm2, Arg1, MRC1/CD206, Alox15) in vehicle or Cp6-treated control and IL-4 (40 ng/mL) stimulated +huAGMO RAW264.7 macrophages. Data were compared with 2-way ANOVA for “stimulation” x “treatment” and subsequent post hoc analysis using a correction of alpha according to Tukey. Statistics as in D. **** *p* < 0.0001.

**Figure 6 cells-10-01081-f006:**
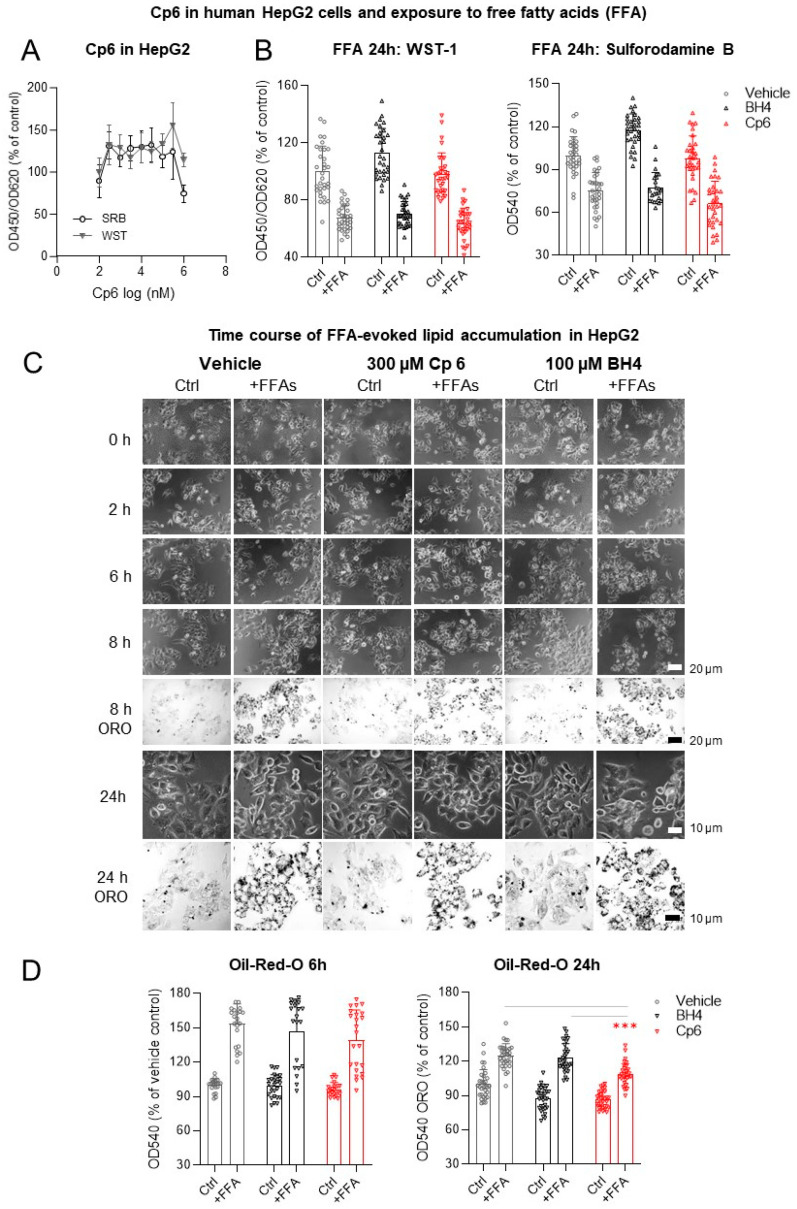
Cp6 does not protect human HepG2 hepatocytes against free fatty acids (FFA) induced toxicity. (**A**) Concentration-dependent effect of Cp6 on cell viability and proliferation as assessed by sulforhodamine B (SRB) and WST-1 assays in human HepG2 hepatocytes. Data are means ± SD of *n* = 3 cultures. (**B**) Effects of Cp6 (300 µM) and BH4 (100 µM) in human HepG2 hepatocytes challenged with FFAs for 24 h. Neither Cp6 nor BH4 were able to prevent FFA-evoked toxicity. Data are the means ± SD 2 × 2 independent experiments per time point. Scatters show pooled replicates. (**C**) Time course of free fatty acid evoked lipid droplet accumulation in human HepG2 cells treated with Cp6 (300 µM), BH4 (100 µM) or vehicle (0.1% DMSO). (**D**) Lipid droplets were quantified per Oil-Red O (ORO) staining and were normalized versus control (vehicle non-FFA) set to 100%. Data show the mean ± SD. The scatter show pooled replicates of 2 × 2 cultures per time point and condition. Data were compared by 2-way ANOVA and subsequent post hoc Šidák; *** *p* < 0.001. Cp6 significantly reduced lipid droplets at 24 h FFA versus vehicle or BH4 groups.

## Data Availability

All data generated in this study are presented within the manuscript or Appendix A.

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
