# Peer review of "AGMO Inhibitor Reduces 3T3-L1 Adipogenesis"

_cells, 2021, doi:10.3390/cells10051081_

Round 1
Reviewer 1 Report
In the manuscript "AGMO inhibitor reduces 3T3-L1 Adipogenesis", the Authors aim to deepen the adipogenic program's knowledge, focusing on the alkylglycerol monooxygenase (AGMO) inhibition. Due to its structural similarity with AGMO cofactor BHT4, a novel AGMO inhibitor has been identified and tested to investigate AGMO's biological role in adipogenesis. First, the authors showed Cp6 pharmacological potency and selectivity over two other BH4-dependent enzymes, PAH and FALDH. In addition, since alkylglycerols are essential for adipogenesis, AGMO's differential expression in the liver and because AGMO drives M2-like macrophages polarization, the authors chose 3T3-L1, HepG2, and RAW264.7, respectively, as cellular models. Finally, they assessed the in vivo relevance of obesogenic AGMO expression in high-fat-fed mice.
In the present study, the authors proposed Cp6 as a valuable tool for reducing adipogenesis and prevent liver steatosis.
Main concerns:
- Line 31: "IL-2 driven differentiation of RAW264.7 macrophages towards M2 like macrophages" Here, authors reported IL-2, yet in the manuscript, they have always referred to IL-4. Can the authors explain this discrepancy?
- Line 78: I suggest adding a comma after the word "inhibitors".
- Line 206: authors reported: "…well plate and seeding of 104 HepG2 in 100 μl DMEM supplemented with 10% FCS per.". What does the word "per" stands for?
- Line 319-320: authors reported "AGMO activity was BH4-dependent and reached a maximum at 100-250 μM of the cofactor", however in figure 2B, the maximal AGMO activity seems to be reached with 500uM. Moreover, can the authors confirm that the number reported on the BH4-treatment curve is the concentration of BH4 used? It is unclear why the authors used these concentrations in combination with different ug/mL of microsomal protein. Can the Author elucidate the data present in this figure?
- Why Cp6 concentrations are expressed as 10exp nM (figure 1, panels D1, D2, D3, and other figures) while in the manuscript is always reported in uM? I suggest harmonizing the Cp6 concentration between graphs and text to make everything more straightforward.
- In figure 1, panel D4, the statistical significance of these data is missing. Please add asterisks and a short description in the figure legend.
- Can the authors speculate on the different effects of Cp6 on low or high-density cells?
Moreover, because of the data in this figure (2B), can the author suppose that the Cp6 effect acts at the beginning of adipocyte differentiation?
- Line 375: the authors show that 3T3-L1 proliferation and differentiation are blunted under Cp6 treatment. However, no data reported the proliferation rate of these cells during adipogenesis in vehicle and Cp6 treated cells. In addition to bright field images, I suggest performing a proliferation curve during adipogenesis (or at least at day -2, 0, 3, 6, and 10), e.g., cell count analysis to highlight MCE and to better specify at which point during differentiation Cp6 treatment causes a decrease in proliferation (in cell number).
To have proper differentiation, the cells must reach confluence. If the Cp6 effect on proliferation occurs before confluence, it must be ensured that cp6-treated cells are induced only after the confluence is reached.
- Line 393: I suggest changing the expression "maintenance of the expression" with "unaltered expression."
- In figure 3, panel D, it is unclear why AGMO expression is raised under Cp6 treatment compared to the control. Moreover, legends and font dimensions are too small. Furthermore, in the figure legend (Line 428), only asterisks are described in the statistical analysis. Please also include hashtags.
- In figure 4B bottom, histogram colors seem different from cell cycle analysis in panel 4A bottom. E.g., the red G2 phase is predominant in Cp6 treated cells + IDI, while the histogram shows a small red bar. Can the Authors explain this misleading information?
- Line 446: you mentioned that concentrations between 0.3 and 1mM were used, yet in figure 4, panel A bottom only 300uM concentration is displayed.
- Line 452: Remove the following paragraph “The drawing shows the cell cycle and major cyclins involved in transitions” from the figure legend. It is redundant and already explained in the paragraph above.
- In figure 5, the A and B panels are missing. Add "A" and "B" on "RAW macrophages: QRT-PCR" and "RAW macrophages: AGMO activity", respectively. Moreover, the legend in figure 5B appears to be missing.
- In figure 5A is unclear to me the mouse and human distinction. The gene expression was performed on RAW cells that are a mouse cell line, so why in the graph there are “mouse” and “human” captions with their relative AGMO expression in Sh or +hu?
- Line 478: “mimicked” seems a strong affirmation.
- In paragraph 3.4, the authors stated that AGMO expression was increased in the fatty liver of obese mice but reduced in gonadal white adipose tissue. They explained this misleading data were owing to a predominant WAT hypertrophy rather than adipogenesis. Can the authors clarify if data support this statement or if it is speculation?
- In figure 6D, why the quantification of Oil red O is missing? Moreover, can the authors elucidate if the absorbance data is normalized on the total protein content?
- Why does the experimental numerosity (n) of Oil red O staining in HepG2 are way higher than the same experiment in 3T3-L1?
- Figure S2 shows a panel with eNOS expression, but the mention in the manuscript is missing.
- In paragraph 3.4, authors speculated that high AGMO expression in the liver has a protective effect on FFA-evoked cytotoxicity, yet they denied this hypothesis since HepG2 viability dropped under FFA treatment. So why do you think that AGMO is highly expressed in this tissue? What might be the relevance of using cp6 in vivo with an anti-obesity effect?
- Can the Authors add a reference for the statement in line 557?
- Overall, the figures appear to be in low resolution (or at least in the downloadable version for peer review). Can the authors increase the resolution of the images?
Author Response
In the manuscript "AGMO inhibitor reduces 3T3-L1 Adipogenesis", the Authors aim to deepen the adipogenic program's knowledge, focusing on the alkylglycerol monooxygenase (AGMO) inhibition. Due to its structural similarity with AGMO cofactor BHT4, a novel AGMO inhibitor has been identified and tested to investigate AGMO's biological role in adipogenesis. First, the authors showed Cp6 pharmacological potency and selectivity over two other BH4-dependent enzymes, PAH and FALDH. In addition, since alkylglycerols are essential for adipogenesis, AGMO's differential expression in the liver and because AGMO drives M2-like macrophages polarization, the authors chose 3T3-L1, HepG2, and RAW264.7, respectively, as cellular models. Finally, they assessed the in vivo relevance of obesogenic AGMO expression in high-fat-fed mice.
In the present study, the authors proposed Cp6 as a valuable tool for reducing adipogenesis and prevent liver steatosis.
Thank you for evaluating our manuscript and giving us your helpful suggestions. We have addressed your comments and suggestions below. The responses are in red. Changes in the manuscript are highlighted by WORD track changes. Figures are replaced/updated without track changes.
Main concerns:
Line 31: "IL-2 driven differentiation of RAW264.7 macrophages towards M2 like macrophages" Here, authors reported IL-2, yet in the manuscript, they have always referred to IL-4. Can the authors explain this discrepancy?
It was a mistake. It is IL-4 and is corrected now.
Line 78: I suggest adding a comma after the word "inhibitors".
The comma was added as requested
Line 206: authors reported: "…well plate and seeding of 104 HepG2 in 100 μl DMEM supplemented with 10% FCS per.". What does the word "per" stands for?
"per" was a left over. It is deleted now.
Line 319-320: authors reported "AGMO activity was BH4-dependent and reached a maximum at 100-250 μM of the cofactor", however in figure 2B, the maximal AGMO activity seems to be reached with 500uM.
As suggested, the text was corrected. We now say that a plateau with low variability was reached at 100-250 µM. Variability was high at 500 µM.
Moreover, can the authors confirm that the number reported on the BH4-treatment curve is the concentration of BH4 used?
It is unclear why the authors used these concentrations in combination with different ug/mL of microsomal protein. Can the Author elucidate the data present in this figure?
The numbers at the data points refer to the BH4 concentrations. The labeling is now added. The curve shows AGMO activity in dependence of microsomal protein without or with BH4 at different concentrations. The analysis was done to assess assay conditions for subsequent experiments.
Why Cp6 concentrations are expressed as 10exp nM (figure 1, panels D1, D2, D3, and other figures) while in the manuscript is always reported in uM? I suggest harmonizing the Cp6 concentration between graphs and text to make everything more straightforward.
The figure uses log scaling. The labeling shows the exponent. It is frequently used for enzyme activity assays that test inhibitory compounds. The labeling is now changed to "Cp6 log (nM)" according to a suggestion of Reviewer#3. The labeling is also used for subsequent experiments with Cp6. It distinguishes Cp6-experiments from experiments addressing substrate and cofactor.
In figure 1, panel D4, the statistical significance of these data is missing. Please add asterisks and a short description in the figure legend.
Asterisks are now added and description in the legend
Can the authors speculate on the different effects of Cp6 on low or high-density cells?
Low density cultures are often more sensitive to drug treatments.
Moreover, because of the data in this figure (2B), can the author suppose that the Cp6 effect acts at the beginning of adipocyte differentiation?
The strongest differences were observed at 5-7 days of IDI. The cell cycle analysis shows that cell cycle reentry is retarded, which needs a about 2 cycles to reveal in cell numbers. We cannot answer from our experiments if Cp6 blocks the initiation of cell differentiation.
Line 375: the authors show that 3T3-L1 proliferation and differentiation are blunted under Cp6 treatment. However, no data reported the proliferation rate of these cells during adipogenesis in vehicle and Cp6 treated cells. In addition to bright field images, I suggest performing a proliferation curve during adipogenesis (or at least at day -2, 0, 3, 6, and 10), e.g., cell count analysis to highlight MCE and to better specify at which point during differentiation Cp6 treatment causes a decrease in proliferation (in cell number).
The cell cycle experiments in Figure 4 address proliferation in detail. Sulforhodamine assay and WST assay were done to assess viability and proliferation, the data is shown in Figure 2A. Thank you for your suggestion of future experiments to further specify the effects of Cp6.
To have proper differentiation, the cells must reach confluence. If the Cp6 effect on proliferation occurs before confluence, it must be ensured that cp6-treated cells are induced only after the confluence is reached.
Cell density/confluence was equal before onset of treatment. It would complicate comparability to postpone IDI in Cp6 cultures. Therefore, we used identical schedules for both groups.
Line 393: I suggest changing the expression "maintenance of the expression" with "unaltered expression."
The expression is now changed as suggested.
In figure 3, panel D, it is unclear why AGMO expression is raised under Cp6 treatment compared to the control.
It has been shown with other enzyme inhibitors that expression of the respective enzyme increases. It may be a compensatory mechanism. It occurs for example with cyclooxygenase inhibitors. For Cp6, we do not know the mechanism.
Moreover, legends and font dimensions are too small. Furthermore, in the figure legend (Line 428), only asterisks are described in the statistical analysis. Please also include hashtags.
The figure subpanel is now increased and simplified. The hashtags had shown the effects of IDI. We now focus on the drug effects. Statistically significant differences versus vehicle are shown as asterisks. The legend was adjusted accordingly.
In figure 4B bottom, histogram colors seem different from cell cycle analysis in panel 4A bottom. E.g., the red G2 phase is predominant in Cp6 treated cells + IDI, while the histogram shows a small red bar. Can the Authors explain this misleading information?
We have replaced the exemplary histogram. It was not meant to be misleading.
Line 446: you mentioned that concentrations between 0.3 and 1mM were used, yet in figure 4, panel A bottom only 300uM concentration is displayed.
We show only the lower concentration because the figure was already complex. The legend is now corrected.
Line 452: Remove the following paragraph “The drawing shows the cell cycle and major cyclins involved in transitions” from the figure legend. It is redundant and already explained in the paragraph above.
It is now deleted as suggested.
In figure 5, the A and B panels are missing. Add "A" and "B" on "RAW macrophages: QRT-PCR" and "RAW macrophages: AGMO activity", respectively. Moreover, the legend in figure 5B appears to be missing.
The legend was rechecked. "A" and "B" were added. B was not missing in the legend.
In figure 5A is unclear to me the mouse and human distinction. The gene expression was performed on RAW cells that are a mouse cell line, so why in the graph there are “mouse” and “human” captions with their relative AGMO expression in Sh or +hu?
It is sh (small hairpin) AGMO (i.e. knockdown of the endogenous mouse AGMO = shAGMO), and overexpression of human AGMO in RAW cells (+huAGMO). The details of the cell lines are described in Suppl. Table 2 and were published previously (Watschinger et al. PNAS 2015). The labeling is shAGMO = knockdown of endogenous mouse AGMO, +huAGMO = overexpression of human AGMO in addition to endogenous mouse AGMO, shLUC control cell line with sh-vector expressing luciferase.
Line 478: “mimicked” seems a strong affirmation.
The sentence is reworded now.
In paragraph 3.4, the authors stated that AGMO expression was increased in the fatty liver of obese mice but reduced in gonadal white adipose tissue. They explained this misleading data were owing to a predominant WAT hypertrophy rather than adipogenesis. Can the authors clarify if data support this statement or if it is speculation?
We have added/changed a paragraph in the discussion. Presently, it is a hypothesis.
In figure 6D, why the quantification of Oil red O is missing? Moreover, can the authors elucidate if the absorbance data is normalized on the total protein content?
Thank you for pointing out that we had not yet quantified all experiments. We have rearranged the figure and now also show the quantification of Oil Red at 6h and at 24h and re-analyzed the data. The results at 24h was significant now for 24h. The legend and the text are reworded/rewritten accordingly.
Why does the experimental numerosity (n) of Oil red O staining in HepG2 are way higher than the same experiment in 3T3-L1?
We now also show more replicates for Oil Red O in adipocytes. Results in adipocytes are from 3x2 independent experiments (the previous graphs had shown the means per experiment), in hepatocytes 2x2 per time point.
Figure S2 shows a panel with eNOS expression, but the mention in the manuscript is missing.
eNOS is now added in the text.
In paragraph 3.4, authors speculated that high AGMO expression in the liver has a protective effect on FFA-evoked cytotoxicity, yet they denied this hypothesis since HepG2 viability dropped under FFA treatment. So why do you think that AGMO is highly expressed in this tissue?
We have re-analyzed HepG2 Oil-Red data. The analysis revealed that Cp6 significantly reduced the FFA evoked increase of Oil-Red staining of lipid droplets at 24h, but it did not affect the FFA evoked drop of WST or SRB. Hence, Cp6 had some protective effect in terms of lipid accumulation in this in vitro model for liver steatosis. The text is rewritten accordingly.
AGMO is needed in the liver for metabolism of alkylglycerol ethers. The observed increase under HFD suggests an on-demand upregulation.
What might be the relevance of using cp6 in vivo with an anti-obesity effect?
We have toned down the conclusion and changed the discussion in terms of metabolic diseases/obesity.
Can the Authors add a reference for the statement in line 557?
A reference was added.
Overall, the figures appear to be in low resolution (or at least in the downloadable version for peer review). Can the authors increase the resolution of the images?
The resolution is 300 dpi, which is the recommended resolution. There are no pixels even on maximum zoom-in. We shall provide higher resolution if requested from the Journal.
Reviewer 2 Report
Summary
Fischer C et al. identify Cp6 as a compound that inhibits AGMO activity with an IC50 of about 50 µM. They show that differentiation of 3T3-L1 fibroblast to adipocytes upregulates AGMO. Moreover, treatment of 3T3-L1 fibroblast with 1000 µM Cp6 inhibits differentiation into adipocytes. In parallel, 1000 µM Cp6 reduced viability and metabolic activity of low density 3T3-L1 fibroblast and cultured macrophages (RAW cells), suggesting that such dose may be toxic. Treatment of 3T3-L1 cells with 300 µM Cp6 reduced cell cycle progression as shown by more <G1 cells and an more cells in the S-phase as well as higher cyclin D expression. In RAW cells, 100 µM Cp6 reduced IL-4 induced M2 polarization but increased LPS/IFNƴ-induced M1 polarization. While the identification of Cp6 as an AGMO inhibitor is of interest, experiments performed in 3T3-L1 adipocytes need some clarification. Also the drawn conclusion that “pharmacological AGMO inhibition might be useful to attenuate obesity” is not well supported by the presented data.
Broad comments
As outlined above, authors perform experiments in 3T3-L1 adipocytes (300, 1000 µM) and RAW cells (100 µM) at different Cp6 concentrations. Why did authors decide to choose different concentrations for different experiments? As a Cp6 concentration of 1000 µM may increase cytotoxicity and decrease viability in some cells, experiments showing a negative effect of Cp6 on adipocytes differentiation need to be performed with lower concentration than 1000 µM. Moreover, the same concentration of Cp6 should be chosen for experiments, as least within the same cell line. In the current state, it is hard to compare data presented in Figure 3 and 4, as different concentrations were used.
As mentioned by the authors (line 83), formation of new adipocytes from precursor cells (adipogenesis) may result in a higher number of small adipocytes. Moreover, small adipocytes may be metabolically more favorable compared to large/hypertrophic adipocytes (line 555). In support of such notion, activation of PPARƴ by thiazolidinedione results in an increase in the number of small adipocytes and improves glucose metabolism. Herein, authors show that Cp6 impairs adipogenesis as revealed by reduced lipid droplet accumulation and reduced PPARƴ expression. Accordingly, one can speculate that treatment with Cp6 may lead to less small adipocytes but rather an increase in the number of hypertrophic cells, which may have negative effects on metabolism. In line, the observation of reduced AGMO expression found in white adipose tissue of HFD-fed mice may suggest that reduced AGMO is a marker of de-differentiation of adipocytes. Therefore, inhibition of AGMO using Cp6 may have negative consequences on obesity and associated diseases and the drawn conclusion that AMGO inhibition might be useful to attenuate obesity may be reconsidered.
Specific comments
Please change IL-2 to IL-4 in the abstract (line 32)
What kind of high fat diet was used? Please add such information to the Materials and Methods section.
Were chow and high fat diet fed mice at the same age at scarification? Please add/clarify statement regarding age of mice in the Materials and Methods section.
Labeling (A and B) is missing in Figure 5.
The statement on line 554 may be adapted to “WAT expansion results from…”
Author Response
Fischer C et al. identify Cp6 as a compound that inhibits AGMO activity with an IC50 of about 50 µM. They show that differentiation of 3T3-L1 fibroblast to adipocytes upregulates AGMO. Moreover, treatment of 3T3-L1 fibroblast with 1000 µM Cp6 inhibits differentiation into adipocytes. In parallel, 1000 µM Cp6 reduced viability and metabolic activity of low density 3T3-L1 fibroblast and cultured macrophages (RAW cells), suggesting that such dose may be toxic. Treatment of 3T3-L1 cells with 300 µM Cp6 reduced cell cycle progression as shown by more <G1 cells and an more cells in the S-phase as well as higher cyclin D expression. In RAW cells, 100 µM Cp6 reduced IL-4 induced M2 polarization but increased LPS/IFNƴ-induced M1 polarization. While the identification of Cp6 as an AGMO inhibitor is of interest, experiments performed in 3T3-L1 adipocytes need some clarification. Also the drawn conclusion that “pharmacological AGMO inhibition might be useful to attenuate obesity” is not well supported by the presented data.
Thank you for evaluation of our manuscript and giving your helpful suggestions. We have addressed your criticisms and our responses to your comments are detailed below.
We have reworded the conclusion as suggested. We have added experiments with 300 µM in adipocytes (now Suppl. Fig.2) and re-analyzed Oil-Red O time points of HepG2 cells, which revealed a significant reduction of ORO at 24h of FFA exposure. The discussion/conclusion is adjusted accordingly. In the Abstract we now say that pharmacologic AGMO inhibition may affect lipid storage.
Broad comments
As outlined above, authors perform experiments in 3T3-L1 adipocytes (300, 1000 µM) and RAW cells (100 µM) at different Cp6 concentrations. Why did authors decide to choose different concentrations for different experiments? As a Cp6 concentration of 1000 µM may increase cytotoxicity and decrease viability in some cells, experiments showing a negative effect of Cp6 on adipocytes differentiation need to be performed with lower concentration than 1000 µM. Moreover, the same concentration of Cp6 should be chosen for experiments, as least within the same cell line. In the current state, it is hard to compare data presented in Figure 3 and 4, as different concentrations were used.
We started with 3T3 cells with 1 mM and learnt from these experiments that 300 µM would have been enough, and we used 300 µM in subsequent experiments. We have now added a repetition of the 3T3 differentiation experiment with 300 µM, which is now presented as Suppl. Fig. 2. Cp6 300 µM still reduced the switch of preadipocyte to adipocyte marker expression and significantly reduced oil-red.
As mentioned by the authors (line 83), formation of new adipocytes from precursor cells (adipogenesis) may result in a higher number of small adipocytes. Moreover, small adipocytes may be metabolically more favorable compared to large/hypertrophic adipocytes (line 555).
In support of such notion, activation of PPARƴ by thiazolidinedione results in an increase in the number of small adipocytes and improves glucose metabolism. Herein, authors show that Cp6 impairs adipogenesis as revealed by reduced lipid droplet accumulation and reduced PPARƴ expression. Accordingly, one can speculate that treatment with Cp6 may lead to less small adipocytes but rather an increase in the number of hypertrophic cells, which may have negative effects on metabolism. In line, the observation of reduced AGMO expression found in white adipose tissue of HFD-fed mice may suggest that reduced AGMO is a marker of de-differentiation of adipocytes. Therefore, inhibition of AGMO using Cp6 may have negative consequences on obesity and associated diseases and the drawn conclusion that AMGO inhibition might be useful to attenuate obesity may be reconsidered.
Thank you for suggestion of these Discussions. We agree, that the in vivo outcome of AGMO inhibition may also be negative. We have added the text, where is fitted into the Discussion and changed the conclusion as suggested. Indeed, there appears to be an enlargement of Cp6 treated cells at 300 µM, suggesting that is might cause adipocyte hypertrophy.
According to suggestions of Reviewer #1 we have re-quantified time-point data for hepatocytes and results now reveal that Cp6 reduced the FFA-evoked increase of Oil-Red staining suggesting a reduction of lipid droplets. Results and Discussion are changed accordingly.
Specific comments
Please change IL-2 to IL-4 in the abstract (line 32)
Thank you, it was a mistake
What kind of high fat diet was used? Please add such information to the Materials and Methods section.
It was a Western type diet. We have added the diet manufacturer (https://www.ssniff.de/) and product number in the Methods.
Were chow and high fat diet fed mice at the same age at scarification? Please add/clarify statement regarding age of mice in the Materials and Methods section.
The age description was unclear and is corrected now. Mice were 22-24 weeks old at onset of HFD or control diet and were sacrificed after 16 weeks of continuous diet i.e. 40 weeks old. One control mouse used as replacement was 50 weeks old at sacrifice. This mouse was close to the mean of the others.
Labeling (A and B) is missing in Figure 5.
Added now
The statement on line 554 may be adapted to “WAT expansion results from…”
Changed as suggested.
Reviewer 3 Report
The authors present a manuscript exploring the inhibition effect of a compound (Cp6) over AGMO during Adipogenesis. The experimental design and presentation are appropriate. However few places need some clarifications.
Comments
- Fig 1 graphs in D Y-axis 10exp nM. Please rewrite the full name or represent it in a different way. “Cp6 concentration versus effect analysis (Fig. 1; D1) showed a concentration-dependent sigmoidal inhibition of AGMO with half-maximum inhibition at about 50 µM (IC50 ranging from 26 to 100 µM)The IC50 for phenylalanine hydroxylase (PAH), which is one of the BH4-dependent hydroxylases, was about 5fold higher (IC50 360-430 µM, Fig.1; D3), whereas inhibition of NOS activity in RAW264.7 macrophages (mainly NOS3) required about 20fold higher concentrations of Cp6 (IC50 1.9-5.3 mM, Fig. 1; D2)” the y-axis still in the same range for all the different IC50 range from 20uM to 5 mM. Please check.
- Please provide the corresponding IC50 (con vs inhibition) graphs.
- Minor :
Line 29 “were”
Line 99 “liver tissue”
Author Response
The authors present a manuscript exploring the inhibition effect of a compound (Cp6) over AGMO during Adipogenesis. The experimental design and presentation are appropriate. However few places need some clarifications.
Thank you for evaluation of our manuscript. We have addressed your comments below. The responses are in red.
Comments
Fig 1 graphs in D Y-axis 10exp nM. Please rewrite the full name or represent it in a different way. “Cp6 concentration versus effect analysis (Fig. 1; D1) showed a concentration-dependent sigmoidal inhibition of AGMO with half-maximum inhibition at about 50 μM (IC50 ranging from 26 to 100 μM). The IC50 for phenylalanine hydroxylase (PAH), which is one of the BH4-dependent hydroxylases, was about 5fold higher (IC50 360-430 μM, Fig.1; D3), whereas inhibition of NOS activity in RAW264.7 macrophages (mainly NOS3) required about 20fold higher concentrations of Cp6 (IC50 1.9-5.3 mM, Fig. 1; D2)” the y-axis still in the same range for all the different IC50 range from 20uM to 5 mM. Please check.
The x-axes are now relabeled as Cp6 log (nM) which is a frequently used label for enzyme inhibition experiments. The Y-axes have not the same ranges. The x-axis scaling are equal to easily reveal the differences, for example right shift of the curve of NOS inhibition versus AGMO inhibition.
- Please provide the corresponding IC50 (con vs inhibition) graphs.
The graphs show Cp6 concentrations (X-axis) versus enzyme activity (Y-axis) to reveal effects of BH4 on enzyme active. Percent inhibition is used for NOS. The IC50 provides 50% inhibition, The IC50 range is given in the graphs which are presented. It is unclear which type of additional graphs are asked for.
- Minor : Line 29 “were”
Line 99 “liver tissue”
The missing words were added as suggested.
Round 2
Reviewer 2 Report
The authors adequately adressed my concerns. I have no further comments.
Reviewer 3 Report
The authors have addressed the concern and modify accordingly.